# Biosynthesis and Pharmacological Activities of Flavonoids, Triterpene Saponins and Polysaccharides Derived from *Astragalus membranaceus*

**DOI:** 10.3390/molecules28135018

**Published:** 2023-06-27

**Authors:** Miaoyin Dong, Jinjuan Li, Delong Yang, Mengfei Li, Jianhe Wei

**Affiliations:** 1College of Life Science and Technology, Gansu Agricultural University, Lanzhou 730070, China; dongmy@gsau.edu.cn (M.D.); yangdl@gsau.edu.cn (D.Y.); 2State Key Laboratory of Arid Land Crop Science, Gansu Agricultural University, Lanzhou 730070, China; 3Institute of Agricultural Quality Standards and Testing Technology, Gansu Academy of Agricultural Sciences, Lanzhou 730070, China; lijinjuan850506@163.com; 4Agronomy College, Gansu Agricultural University, Lanzhou 730070, China; 5Institute of Medicinal Plant Development, Chinese Academy of Medical Sciences & Peking Union Medical College, Beijing 100193, China

**Keywords:** *Astragalus membranaceus*, flavonoids, triterpene saponins, polysaccharides, biosynthesis, pharmacological activity

## Abstract

*Astragalus membranaceus* (*A. membranaceus*), a well-known traditional herbal medicine, has been widely used in ailments for more than 2000 years. The main bioactive compounds including flavonoids, triterpene saponins and polysaccharides obtained from *A. membranaceus* have shown a wide range of biological activities and pharmacological effects. These bioactive compounds have a significant role in protecting the liver, immunomodulation, anticancer, antidiabetic, antiviral, antiinflammatory, antioxidant and anti-cardiovascular activities. The flavonoids are initially synthesized through the phenylpropanoid pathway, followed by catalysis with corresponding enzymes, while the triterpenoid saponins, especially astragalosides, are synthesized through the universal upstream pathways of mevalonate (MVA) and methylerythritol phosphate (MEP), and the downstream pathway of triterpenoid skeleton formation and modification. Moreover, the *Astragalus* polysaccharide (APS) possesses multiple pharmacological activities. In this review, we comprehensively discussed the biosynthesis pathway of flavonoids and triterpenoid saponins, and the structural features of polysaccharides in *A. membranaceus*. We further systematically summarized the pharmacological effects of bioactive ingredients in *A. membranaceus*, which laid the foundation for the development of clinical candidate agents. Finally, we proposed potential strategies of heterologous biosynthesis to improve the industrialized production and sustainable supply of natural products with pharmacological activities from *A. membranaceus*, thereby providing an important guide for their future development trend.

## 1. Introduction

*Astragalus membranaceus* (*A. membranaceus*) mainly consists of *Astragalus membranaceus* (Fisch.) Bge. and *Astragalus membranaceus* (Fisch.) Bge. var. *mongholicus* (Bge.) Hsiao is a perennial herbaceous plant of the legume family [1]. The dried root of *A. membranaceus*, known as Radix Astragali or “Huangqi” in Chinese, has been one of the most commonly used traditional herbal medicines for more than 2000 years in China and other Asian countries, with hepatoprotective, tonic and expectorant properties and as a diuretic [2,3]. The medicinal efficacy of *A. membranaceus* was firstly documented in the Divine Farmer’s Materia Medica (Shennong Bencao Jing), the earliest extant pharmaceutical monograph in China. Furthermore, modern pharmacological studies have found that *A. membranaceus* is clinically beneficial for the treatment of respiratory, immunological, cardiovascular and hepatic diseases [4,5,6]. In addition, *A. membranaceus* has also commonly been used as a dietary supplement and additive in European and American countries; in particular, the United States has classified *A. membranaceus* as an over-the-counter dietary supplement that can be sold at health food markets [7,8].

*A. membranaceus* can widely synthesize a variety of bioactive components as secondary metabolites, such as flavonoids, triterpene saponins and polysaccharides, which determine the quality of the medicinal materials to a large extent [9]. Among them, flavonoids are the most abundant and ubiquitous secondary metabolites distributed in *A. membranaceus* tissues, mainly in the roots, stems, leaves, flowers, fruits and seeds. Although flavonoids have various chemical structures, they share the same structural skeleton in the early stages of biosynthesis, which mainly contains three rings (C6-C3-C6) [10,11]. To date, more than 52 flavonoid components, including flavones, isoflavones, flavanones, flavonols, chalcones and anthocyanidins, have been identified in *A. membranaceus* [1]. In nature, triterpenes and their saponins are the second largest secondary metabolites with a total of more than 20,000, which are widely distributed in higher plants, dicotyledons, monocotyledons, fungi, pteridophytes and marine organisms [12], while the tetracyclic and pentacyclic triterpenoids are common triterpene saponins compounds in *A. membranaceus*. The astragalosides, important tetracyclic triterpenoids compounds, have a wide range of biological activities and important pharmacological effects [13]. Astragaloside and calycosin-7-*O*-*β*-D-glucoside (CG) are considered to be the most important bioactive ingredients that belong to the triterpene saponins and flavonoids in *A. membranaceus*, respectively, and are often used as “marker components” in the Chinese Pharmacopoeia [14], and astragaloside is also recorded in the European Pharmacopoeia and British Pharmacopoeia [15,16].

Currently, the interest of numerous scholars around the world has been attracted to *A. membranaceus* due to its potential pharmacological activities and therapeutic effects. Thus, it is imperative to solve the challenges of the resources and production of *A. membranaceus*. In particular, the demand for *A. membranaceus* is rapidly growing in the pharmaceutical, nutraceutical, food and cosmetics industries. Nevertheless, with the recklessly extensive excavation of medicinal resources, wild *A. membranaceus* has become an endangered species [17]. Therefore, we have to comprehensively understand the biosynthetic pathways and corresponding key genes involved in flavonoids’, triterpene saponins’ and polysaccharides’ biosynthesis in *A. membranaceus*, which have provided great guidance for the medicinal plant breeding and novelty drug exploration derived from *A. membranaceus*.

In this review, we systematically summarized the biosynthetic pathways and involved key enzymes, and the bioactivity of flavonoids, triterpene saponins and polysaccharides in *A. membranaceus*. In addition, we proposed potential strategies of heterologous biosynthesis to improve the industrialized production and sustainable supply of active components with pharmacological activities in *A. membranaceus*, aiming to offer new insights into the exploration and biomanufacturing of natural products.

## 2. Biosynthesis of Flavonoids

Flavonoids are ubiquitous in *Astragalus* tissues, including the roots, stems, leaves, flowers, fruits and seeds. The biosynthetic pathway of flavonoids, especially its subclass isoflavonoids, has been widely studied in *A. membranaceus* and partially elucidated [18,19]. The upstream pathway of flavonoids biosynthesis starts from the L-phenylalanine involved in the phenylpropanoid pathway (Figure 1), which is necessary for the growth of plants and the result of the long-term adaptation of plants to natural conditions [20]. Furthermore, the phenylpropane pathway is one of the main universal routes used to synthesize secondary metabolites in plants [21].

The first committed step is regulated by the phenylalanine ammonia lyase (PAL), which catalyzes the deamination of L-phenylalanine and produces the cinnamic acid (Figure 1). PAL is an important rate-limiting enzyme in phenylpropane metabolism, which is largely expanded and experienced in tandem duplication events in *A. membranaceus* [22]. Liu et al. isolated an *AmPAL1* (GenBank No. AY986506) gene from the *A. membranaceus*, which was significantly induced by mechanical wounding, UV irradiation and white light irradiation, and expressed universally in various organs [23]. In addition, they found that the contents of flavonoids and quercetin in *A. membranaceus* of different ages were closely related with the PAL enzymatic activity.

Subsequently, *p*-coumaric acid is generated through hydroxylation at the C-4 positions by a cytochrome P450 monooxygenase of cinnamate 4-hydroxylase (C4H) [24]. Next, the 4-coumarate-CoA ligase (4CL) catalyzes the conversion of *p*-coumaric acid into *p*-coumaroyl-CoA (Figure 1), which acts as an important precursor for various synthetic derivatives involved in the phenylpropanoid pathway, including lignans and flavonoids [25].

The biosynthesis of all flavonoids in *Astragalus* starts from the *p*-coumaroyl-CoA and is catalyzed by chalcone synthase/chalcone reductase (CHS/CHR), which catalyzes the condensation of *p*-coumaroyl-CoA and three molecules of malonyl-CoA to form a chalcone intermediate, such as isoliquiritigenin or naringenin chalcone [26]. In the flavanones biosynthetic pathway, the chalcone isomerase (CHI) catalyzes chalcone substrates to produce flavanones (Figure 1), which are transformed to various isoflavones, flavones, flavonols and dihydorflavonols by the respective enzymes [27]. For isoflavones synthesis, the CHI catalyzes the cyclization of isoliquiritigenin and generates liquiritigenin involved in downstream enzymatic reactions for ultimate isoflavone compounds’ formation in *Astragalus* [28]. Subsequently, the daidzein is synthesized under the catalyzation of isoflavone synthase (IFS), which converts the flavanones to their corresponding isoflavones through a 2, 3 aryl ring migration. Isoflavones are synthesized almost exclusively in leguminous plants under the catalysis of IFS, which is deficient in most other plant species [29].

Finally, isoflavone 3′-hydroxylase (I3′H) catalyzes the hydroxylation reaction at the 3′-position of formononetin to produce calycosin (Figure 1), which is continuously transformed into the CG under calycosin 7-O-glucosyltransferase (UCGT) [30]. The final step of UGT-catalyzed glycosylation in the isoflavones’ biosynthesis pathway promotes the stability, solubility and specific bioactivity of isoflavones’ secondary metabolites and adaptation to the harsh environment of *A. membranaceus* plants [31,32,33]. In a recent study, two 7-*O*-glycosyltransferases of *AmUCGT* (c303354, GenBank No. MN241498) and *AmUFGT* (c778119, GenBank No. ON375915), which catalyze calycosin and formononetin, respectively, were predicted by phylogenetic analysis [21]. Kim et al. found that the transcript levels of the main genes (*AmPAL*, *AmC4H*, *Am4CL*, *AmCHR*, *AmCHS*, *AmCHI*, *AmIFS*, *AmI3′H and AmUCGT*) related to the biosynthesis of calycosin-7-*O*-*β*-D-glucoside and calycosin were higher in the flower than those of other organs (leaf, stem and root) in *A. membranaceus*; thus, they speculated that the accumulation of calycosin-7-*O*-*β*-D-glucoside in roots might originate from the calycosin in the stem and leaf [25]. In addition, most of the UGTs’ genes family were distributed on chromosomes as gene clusters in *A. membranaceus* [22]. The key enzymes of C4H, IFS and I3′H involved in isoflavone biosynthesis belong to the CYP450 superfamily, which are located on the nine chromosomes and have the largest enrichment (46, 19%) on chromosome 5 in *A. membranaceus* [22].

## 3. Biosynthesis of Triterpenoid Saponins

Triterpenoid saponins are a group of commonly natural products with a diverse structure and important bioactivities [34]. Triterpenoids are formed by the condensation of triterpenoid saponins coupled with sugar or other chemical groups, while the tetracyclic triterpenoids (Astragalosides) and pentacyclic triterpenoids (Oleanolic acid) are common compounds in *A. membranaceus*. Biosynthesis of triterpenoid saponins is a complex multi-step process, mainly including precursors synthesis, chain elongation, isomerization, cyclization, chain coupling, epoxide protonation and glycosylation [35]. In general, the biosynthesis of triterpenoid can be divided into three stages: (1) the synthesis of upstream precursors, such as isopentenyl diphosphate (IPP) and dimethylallyl diphosphate (DMAPP); (2) the synthesis of the carbocyclic skeleton; and (3) the formation of different kinds of triterpenoid saponins [34]. Triterpenoid saponins are synthesized in plants through the universal upstream pathways of the mevalonate (MVA) and methylerythritol phosphate (MEP) pathways and the downstream pathway of triterpenoid skeleton formation and modification (Figure 2).

### 3.1. Upstream Biosynthesis Pathway

Firstly, the upstream precursor of IPP is synthesized from the initial substrate of acetyl CoA via a six-step condensation reaction in the MVA pathway [12,36]. The MEP pathway also synthesizes the IPP by utilizing the glyceraldehydes 3-phosphate and pyruvate as starting substrates through a seven-step reaction. In addition, the MEP pathway produces polyterpenes along with the chloroplast-bound isoprenoids (b-carotene, prenyl chains of plastoquinone and chlorophylls) in the plastid [37]. Nevertheless, the MVA pathway is well-clarified and plays a dominant role in the biosynthesis of triterpenoid saponins, which mainly synthesizes the isoprenoids, sterols, ubiquinones and sesquiterpenes in the cytoplasm and mitochondria [38]. The upstream precursor of IPP and DMAPP are interconverted under the catalyzation of isopentenyl diphosphate isomerase (IDI).

### 3.2. Downstream Biosynthesis Pathway

Subsequently, the geranyl diphosphate (GPP, C10) is synthesized from the substrate of IPP and DMAPP through the condensation reaction by geranyl pyrophosphate synthase (GPS), and then converted to farnesyl diphosphate (FPP, C15) by adding the second IPP unit under the action of the farnesyl diphosphate synthase (FPS) [39]. In the stage of triterpenoid skeleton formation, the downstream precursor of squalene is formed from the condensation of two FPPs by squalene synthase (SS), which is considered a rate-limiting enzyme in the triterpenoids’ biosynthesis [40,41]. Next, the squalene is oxidized into 2,3-oxidosqualene under the catalyzation of squalene epoxidase (SE), serving as the key precursor of triterpenoids saponins [12].

Moreover, the most important step of tetracyclic triterpenoids skeletons’ formation is the cyclization of 2,3-oxidosqualene and generation of cycloartenol, which is catalyzed by the cycloartenol synthase (CAS) that belongs to the oxidosqualene cyclase (OSCs) family [42]. Duan et al. (2023) identified a cycloartenol synthase of *Am*CAS1 from *A. membranaceus* through the in vivo (in yeast) and in vitro functional identification based on the guidance of transcriptome and phylogenetic analysis [43]. They found that this enzyme catalyzes the cyclization of 2,3-oxidosqualene into cycloartenol *A. membranaceus.* The cycloartenol is derived from the C-20 protosteryl cation, which is generated from the chair-boat-chair (CBC) conformation of 2,3-Oxidosqualene after folding. In the process of cyclization, the epoxide group of 2,3-Oxidosqualene is initially protonated, which triggers a carbocationic cyclization and rearrangement cascade, and forms diverse triterpene skeletons after deprotonation reactions [44].

On the other hand, the 2,3-Oxidosqualene substrate is cyclized to the chair-chair-chair (CCC) conformation and gives the tetracyclic dammarenyl cation. This cation may undergo further conversion to the pentacyclic triterpenoids by β-amyrin synthases (β-AS) and lupeol synthases (LUS), including lupeol, α-amyrin and β-amyrin [45]. Thus, the OSCs is an important enzyme and branch point to synthesize tetracyclic triterpenoids and pentacyclic triterpenoids. The OSCs mainly includes CAS, α-amyrin synthase (α-AS), β-AS and LUS in plants, which catalyze the cyclization of 2,3-Oxidosqualene and generate more than 100 variations of the triterpenoid and sterol skeleton [45,46]. In addition, the flexible and changeable core site of OSCs maybe causes the various structures of triterpenoid saponins in that the catalytic properties of OSCs mainly rely on its amino acid sequence [47]. Two OSC genes, *AmOSC2* and *AmOSC3*, were identified from *A. membranaceus*, and their functions were studied by heterologous expression in tobacco and yeast [48]. *AmOSC2* is a β-amyrin synthase that is associated with the synthesis of β-amyrin and soyasaponins in vivo and shows higher expression activity in underground parts, while *AmOSC3* is a cycloartenol synthase, which is closely related to the production of cycloartane-type astragalosides and cycloartenol.

Then the triterpenoid backbone undergoes diverse structural modifications (such as oxidation, glycosylation and substitution), mainly performed by the cytochrome P450-dependent monooxygenases, glycosyltransferases (UGTs) and other enzymes [49]. Chen et al. found that the key genes involved in triterpenoids biosynthesis, such as *OSCs*, *CYP450* and *UGT*, were closely linked and clustered in the genome of *A. membranaceus*, which implied that there was collaborative expression of genes during the triterpenoids’ synthesis [22]. Finally, the diverse triterpenoid saponin compounds are synthesized under the structural modifications by OSCs, CYP450, UGT and other enzymes in *A. membranaceus*.

### 3.3. Biosynthesis Pathway of Astragalosides

Astragalosides, important tetracyclic triterpenoid compounds, are considered to be the main active constituents in *A. membranaceus*, of which cycloastragenol-type glycosides are the most typically bioactive compounds with pharmacological activities. In nature, cycloastragenol-type glycosides are a rare kind of triterpenoid glycoside, which were derived from the cyclization of 2,3-oxidosqualene to cycloartenol with a typical 9,19-cyclopropane moiety. The biosynthesis of cycloastragenol involved a series of oxidations and furan ring formation under the action of oxidases and epoxide hydrolase (Figure 3). There may be an epoxidation reaction in the terminal olefinic bond of cycloartenol, and then the hydroxyl group at the C20 position attacks the epoxide via an epoxide hydrolase, which forms the 20,24-tetrahydrofuran furan ring in the 5-*exo*-tet mechanism [50]. Thus, this 5-member cyclic ether in the side chain is a more common representative structure of astragaloside that belongings to the cycloartane-type saponins. Meanwhile, the cycloastragenol was generated from the cycloartenol by forming hydroxyl groups at the C6, C16 and C25 positions under the action of oxidations [43].

The glycosylation of cycloastragenol catalyzed by a series of UGTs is commonly considered to be the last step in the biosynthesis of astragalosides and its derivatives, resulting in the vast structural diversity and important bioactivity of astragalosides [34]. The different sugar units (glucose or xylose) and numbers of glycosyl moieties (mono-, di-, tri-, or branched sugar chains) are mainly transferred to the 3-OH, 6-OH, 25-OH and 2′-OH position of cycloastragenol by UGTs.

To date, only four glycosyltransferases (AmUGT7, AmUGT8, AmUGT13 and AmUGT15) responsible for cycloastragenol glycosides have been identified in *A. membranaceus* [43,51]. AmUGT15 could catalyze the 3-*O*-xylosylation of cycloastragenol and cycloastragenol-6-*O*-*β*-D-glucoside to form the cycloastragenol-3-*O*-*β*-D-xyloside and astragalosides IV, respectively, while astragalosides IV was assigned as the most important triterpenoid saponin in *A. membranaceus*, which could also be synthesized from the cycloastragenol-3-*O-β*-D-xyloside by the variants A394F of AmUGT8. AmUGT7, a 2′-*O*-glucosyltransferase with regioselectivity towards 2-OH of xylosyl moiety at C3 positions, was able to synthesize astragaloside III. During the biosynthesis of astragalosides, AmUGT13 is mainly responsible for the 25-*O*-glucosylation of astragaloside III and forms astragaloside V [52]. This glycosyltransferase exhibits high regio-specificity and flexibility towards both acceptors and sugar donors. In addition, the other diverse modifications of astragalosides’ structure, including oxygenation, glycosylation and acylation, remain unclear and require further elaboration.

## 4. Structural Features of Polysaccharides

The astragalus polysaccharide (APS) is one of the major bioactive components in *A. membranaceus* with a complex and diverse structure, and possesses multiple pharmacological activities [53]. Although the APS is mainly linked by the glycosidic bonds between different monosaccharides, the deep understanding on accurate components is very limited. Moreover, the types of glycosidic bond and corresponding pharmacological activities of the APS in *A. membranaceus* are also different depending on the basic sources, medicinal part, planting method, place of production and growth year [54]. The polysaccharide components of *A. membranaceus* mainly include heteropolysaccharide, neutral polysaccharide, dextran and acidic polysaccharide, which were linked by the main glycosidic bond types of 1,4-glucose linkage [55].

Most of the APSs isolated and purified from *A. membranaceus* are heteropolysaccharides whose molecular weight ranges from 8.7 to 4800 kDa (Table 1). In addition, the heteropolysaccharides are in water-soluble forms and composed of various monosaccharides, including D-glucose, D-galactose, D-mannose, D-ribose, D-xylose, L-rhamnose, L-rabinose, L-xylose, L-ribose, glucuronic acid (GlcA) and galacturonic acid (GalA) [56], while the dextran extracted from *A. membranaceus* includes α-(1→4) dextran and α-(1→4)(1→6) dextran with water-insoluble and water-soluble forms, respectively [57].

A pure polysaccharide of AX-I-3b (Mw: 7.9 kDa) was extracted with hot water and purified through DEAE-cellulose 52 column and Sephacryl S-400 HR gel column chromatography, which consisted of Ara, Xyl and Glu at the ratio of 10.4, 79.3 and 1.1, respectively, and were linked as follows: →2,3,4)-β-D-Xyl-(1→, →4)-β-D-Ara-(1→, →4)-β-D-Glc-(1→[58]. Jiang et al. obtained a heteropolysaccharide (linked as α and β indicant bonds) of the APS extracted with microwave, which was purified by using ultrafiltration and resin absorbing (DEAE Sepharose FF) [59]. They found that the monosaccharide compositions of the APS were Man, Gal, Fru, Fuc and Xyl. In addition, a cold-water-soluble polysaccharide (Mw: 12.3 kDa) of cAMPs-1A was purified through a DEAE-cellulose 52 anion-exchange column and a Sephadex G-100 column, which consisted of Fuc, Ara, Gal, Glu and Xyl with a molar ratio of 0.01, 0.1, 0.2, 1.0 and 0.1, respectively [60].

**Table 1 molecules-28-05018-t001:** The chemical composition, structural properties and pharmacological activities of polysaccharides derived from the *A. membranaceus*.

Components	Extraction/Purification	Monosaccharide Composition	Structural Information	Molecular Weight (kDa)	Pharmacological Activities	References
AERP1	Hot-water/Sephacryl^®^ S-400 column	Man:Rha:GalA:Glu:Gal:Ara with a molar ratio of 1.00:2.59:12.15:2.60:3.07:4.54	3/5-α-araf-(1→,T-α-araf,→4,6-β-manp-(1→,→3/3,6-β-galp-(1→,→2/2,4-α-rha-(1→,→-4/4,6-α-glcp-(1→,→4-α-galpA-(1→and→4)-6-OMe-α-galpA-(1→	2.01 × 10^3^	Improved diabetes-related cognitive dysfunction	[61]
AERP2	Hot-water/Sephacryl^®^ S-400 column	Glucan	→4/6-α-glcp-(1→ linkage)	2.11	Improved diabetes-related cognitive dysfunction	[61]
APSID3	Hot-water/DEAE Sepharose Fast Flow and Sephacryl S-300 chromatography	Ara:Rha:Gal:Glc with a molar ratio of 2:2:5:6	The minimal repeat unit: one terminal Ara, one 1,5-linked Ara, one 1,3-linked Rha, one 1,3,4-linked Rha, five 1,4-linked GalA and six 1,4-linked GluA	5.8 × 10^2^		[56]
RAP	Boiling water/Buchi Purifier system coupled with a Hiload 26/60 Superdex-200 column	Rha:Ara:Glc:Gal:GalA with a molar ratio of 0.03:1.0:0.3:0.4:0.3	The backbone:1,2,4-linked Rha, α-1,4-linked Glc, α-1,4-linked GalA6Me, β-1,3,6-linked Gal; The side chains: α-T-Ara and α-1,5-linked Ara; The terminal residues: T-linked Ara, T-linked Glc and T-linked Gal.	1.3 × 10^3^	Immunomodulation	[62]
APS-I	Sephadex G-100 column	Glu:Gal:Ara:Rha:GalA with a molar ratio of 1.5:1:5.4:0.08:0.1	1,4-linked D-Glc, 1,2-linked D-Glc, L-Rha, 1,5-linked D-Ara, 1,2,5-linked D-Ara, 1,4-linked D-Ara, D-Gal	5 × 10^2^	Immunomodulation	[63]
APS-II	Sephadex G-100 column	Glu:Gal:Ara:Rha:GalA with a molar ratio of 9:1:1.4:0.04:0.001	1,4-linked α-D-Glc, 1,6-linked α-D-Glc, 1,4,6-linked α-D-Glc, 1,3,4,6-linked α-D-Glc, 1,2-linked α-D-Glc, α-L-Rha, 1,5-linked α-D-Ara, 1,4-linked α-D-Ara, β-D-Gal	10	Immunomodulation	[63]
AX-I-3b	Hot-water/DEAE-cellulose 52 column chromatography and Sephacryl S-400 HR gel column	Ara:Xyl:Glu with a molar ratio of 10.4:79.3:1.1	1,4-linked β-D-Xyl, 1,4-linked β-D-Ara, β-D-Glc	7.9	Immunomodulation and antitumor	[58]
APS	-	Glu:Ara:Xyl:Man:Gal with a molar ratio 95:2.9:0.7:0.7:0.6	-	17.4	Antitumor	[64]
AMA-1-b-PS2		Ara:Fuc:Gal:Glu:Man:Rha:Xyl:GalA:GluA with a molar ratio of 12.8:4.5:25.6:23.6:24.8:5.1:0.7:1.5:1.4.	The backbone: β-D-(1→3) linked galactans		Immunomodulation	[65]
APS-II	DEAE-32 anion-exchange chromatography and Sephacryl S-300 high resolution column chromatography	Xyl:Glu:Ara:Rha:Man:Gal with a molar ratio of 9.2:77.9:1:5.2:4.5:2.2		11.4	Immunomodulation	[66]
AMP	Hot-water/cationic exchange column (Dowex 50 W-x8)	Glu:Ara:Gal with a molar ratio of 91:6.2:2.8		6.9–9.2 × 10^2^	Immunomodulation	[67]
AMon-S	Hot-water/DEAE Sephadex A-25, Con A-Sepharose chromatography, Toyopearl HW60F	Ara:Gal:GalA:Glc with a molar ratio of 18:18:1:1	Structural units: α-Arabino-β-3,6-galactan type	76	Reticuloendothelial system-potentiating activity	[68]
APS	Hot-water/DEAE-Sepharose CL-6B	Glu:Gal:Ara with a molar ratio of 1.75:1.63:1		36	Hepatoprotection	[69]
APS	Hot-water/DEAE-cellulose column and Sephacryl-S400 column	Glc	The repeat units: a (1→4)-linked backbone with a (1→6)-linked branch every 10 residues	20.1	Antioxidant and immunomodulation	[53]
APS2	Boiling water/precipitation with 40% ethanol	Ara		40	Immunomodulation	[70]
APS3	Boiling water/precipitation with 60% ethanol	Rha:Glu:Gal:Ara with a molar ratio of 1:10.8:6.6:12		15.3	Immunomodulation	[70]
APS	Microwave/ultrafiltration and resin absorbing (DEAE Sepharose FF)	Man, Gal, Fru, Fuc, Xyl	Heteropolysaccharide with α and β indicant bonds		Immunomodulation and antiviral	[59]
APS	Boiling water/Sephadex G-100 column	Rha:Xyl:Glc:Gal with a molar ratio of 1:4:5:1.5	linear backbone:1,3-linked β-D-Gal residues with insertion of β-Glc, 1,6-linked α-Gal, 1,5-linked β-Xyl, 1,4-linked β-Gal, β-D-Gal, 1,2-linked α-Rha, 1,2,4-linked α-Rha residues	3.01 × 10^2^	Immunomodulation	[71]
APS	Hot water	Man:Glu:Xyl:Ara:GluA:Rha with a molar ratio of 0.3:12.8:1.6:0.7:1.0:0.6		2.04 × 10^3^	Antiinflammatory	[72]
cAMPs-1A	Cold-water/DEAE-cellulose 52 anion-exchange column and a Sephadex G-100 column	Fuc:Ara:Gal:Glu:Xyl with a molar ratio of 0.01:0.1:0.2:1.0:0.1		12.3	Antitumor	[60]
APS	Hot-water/anion-exchange and gel permeation chromatography	Glc	α-(1→4)-D-glucan, with a single α-D-glucose at the C-6 position every nine residue, on average, along the main chain.	36	Renal protection	[73]
APS	Hot-water/Sephadex G-50 and lyophilized	Ara:Gal:Glu:Man with a molar ratio of 1.00:0.98:3.01:1.52	pyranose ring and α-type glycosidic linkages	2.1	Antitumor	[74]

## 5. Pharmacological Activities of Bioactive Ingredients

*A. membranaceus* has a long history (more than 2000 years) in Chinese herbal medicine for medicinal usage due to its significant bioactivities and pharmacological effects. When compared to other *Astragalus* genus plants (Table 2), *A. membranaceus* exhibited the widest range of pharmacological activities, including anticancer, antidiabetic, antiviral, hepatoprotective, immunomodulatory, antiinflammatory, antioxidant and anti-cardiovascular activities (Figure 4).

### 5.1. Immunomodulatory Effects

Modern pharmacological studies have proven that *A. membranaceus* has immunomodulatory effects by improving the immune system and alleviating the adverse effects of conventional drug treatments [84]. Several studies have proven that the APS (40–400 µg/mL) can efficiently protect the bone marrow mesenchymal stem cells from radiation-induced apoptosis, formaldehyde-induced cytotoxicity and genotoxicity by regulating the relative genes expression, such as B-cell lymphoma-2 (Bcl-2), Bcl-2-associated X (Bax) and xeroderma pigmentosum group A [85,86]. Bao et al. reported that the APS (150 mg/kg) was able to protect the hematopoietic stem cells by improving the bone marrow and hematopoietic stem cell, and increasing the number of CD34^+^ cells, Lin–c-Kit^+^cells and Lin–Sca1^+^c-Kit^+^ cells [87].

In addition, the APS is a regulator for the secretion and production of cytokines, which could improve splenocytes to induce interferon (IFN), produce interleukin 2 (IL2), enhance the secretion of IL3, IL4 and IL6 [9,88,89], and induce the production of IL8, IL10 and IL12 [90,91]. As for the macrophages, the APS of 12.5–100 µg/mL is able to increase the expression and production of NO, IL-1β, TNF-α and IL-6 by activating the MAPK and NF-κB signal pathways [67]. Similarly, Wei et al. found that the APS is able to activate TLR4-related MAPKs signal pathways, including phosphorylated JNK (p-JNK), phosphorylated ERK (p-ERK) and phosphorylated p38 (p-p38), and induce NF-κB translocation and IκB-α degradation [92]. In short, the APS may induce the production of cytokines in RAW264.7 cells by activating the MAPKs and NF-κB signal pathways mediated with TLR4. In addition, Li et al. proved that the APS and deproteinated APS (DP) stimulated the production of NO and the up-regulation of cytokines’ mRNA expression by activating the NF-κB and MAPKs pathways in RAW264.7 cells, while the desulfated AMP (DS) significantly decreased the activation of RAW264.7 and NK cells [67].

Moreover, the important role of APS in Ig is to regulate immunity by secreting IgA, IgG and IgM. In vivo studies indicated that the APS of 8 mg/kg could improve the immunity by promoting the proliferation of T and B cells, and producing a variety of cytokines in cyclophosphamide-induced immunosuppressive mice, such as IgG, IgA, IgM, TNF-α, IL-6, IL-2 and IFN-γ [93]. In previous studies, the APS of 300–1200 mg/kg improved the percentages of CD3^+^CD4^+^ T cells and CD3^+^, and decreased the ratio of CD3^+^CD8^+^/CD3^+^CD4^+^ and the expression of IL-10, IL-6 and TNF-α [94].

In an in vivo animal model of BALB/c mice and Wistar rats, the total flavonoids extracted from *A. membranaceus* promoted the serum hemolysin level and delayed type hypersensitivity, macrophage phagocytic and the immune organ index in mice, while they alleviated mouse ear edema and vascular permeability, and rat paw edema granuloma formation [95].

### 5.2. Anticancer Effects

To date, cancer remains the leading cause of death and significantly influences life expectancy [96]. While *A. membranaceus* has exhibited potential antitumor activities against various tumor types on the basis of inhibiting the tumor growth, migration and invasion [97,98], in recent years, the APS has more commonly been used for the prevention and treatment of various tumors, such as gastric cancer, liver cancer and colon cancer. It was reported that tumor growth and migration were mainly caused by the rapid proliferative capacity of tumor cells [56]. In a murine H22 hepatocarcinoma model, APS treatment (100 and 400 mg/kg) effectively inhibited the growth of a solid tumor transplanted in BALB/c mice, and promoted the expression of TNF-α, IL-2 and IL-12 and decreased the concentration of IL-10 in serum [99]. Liu et al. reported that the oral administration of APS at dosages of 75, 150 and 300 mg/kg significantly inhibited tumor growth and had inhibitory rates of 20.53%, 36.50% and 44.49%, respectively [60]. They also found that the APS protected the immune organs and promoted macrophage pinocytosis in tumor-bearing mice (Figure 5).

The effects of *A. membranaceus* on gastric-cancer-induced mesothelial cell apoptosis were analyzed in a previous study, which reported that the *Astragalus* treatment could partly suppress HMrSV5 cells’ apoptosis induced by the supernatant of MKN45 gastric cancer cells, condensation of chromatin and nuclear fragmentations, and regulate the expressions of Bax and Bcl-2 in the HMrSV5 cells [100]. Flavonoids and isoflavonoids are the most important secondary metabolites in a plant, with more than 8000 compounds to date [29], which have been proven to impede the growth of cancer cells by the modulation of apoptosis, and by inhibiting the DNA topoisomerase and tyrosine kinase activities [101]. In a study by Zhang et al., the calycosin inhibited the pancreatic cancer cell growth through the induction of cell cycle arrest induced by p21^Waf1/Cip1^ and apoptosis in a caspase-dependent manner, which also promoted the migration of MIA PaCa-2 cells via the epithelial–mesenchymal transition and by activation of a matrix metalloproteinase [102].

In addition, astragaloside-IV (AS-IV) remarkably inhibited the growth of a tumor in vivo based on immune enhancement activity by inducing the CTLs activity and inhibiting the Tregs expression in an orthotopic lung cancer model of C57BL/6 mice [103]. It was reported that AS-IV inhibited the migration and invasiveness of hepatocellular carcinoma cells by significantly down-regulating the expression of lncRNA-ATB in a time- and dose-dependent manner by blocking the signaling pathway of IL-11/STAT3 [97]. *Astragalus saponins* (AST) promoted the apoptosis of HT-29 colon cancer cells in a caspase 3- and polymerase-dependent manner, and inhibited cell proliferation by regulating the cell cycle of the S and G2/M phase, with concomitant inhibition of p21 expression and cyclin-dependent kinase activity. In an in vivo study, the antitumorigenic effects of AST were similar to the conventional chemotherapeutic drug 5-fluorouracil (5-FU), such as the reduction in tumor volume and the pro-apoptotic and antiproliferative effects in a mice xenograft [104]. Therefore, AST could be used for tumor therapy as an effective chemotherapeutic agent, or combined with other orthodox chemical drugs in order to alleviate the systemic side effects of toxic chemotherapeutic compounds.

### 5.3. Antiinflammatory and Antioxidant Effects

Numerous studies have proven that the most bioactive ingredients derived from *A. membranaceus* have antiinflammatory and antioxidant effects, and thus have been widely used in clinic. Adesso et al. proved that the extract derived from *A. membranaceus* reduces the lipopolysaccharide (LPS, derived from *E. coli*) plus interferon-γ-induced inflammatory response, and decreases the expression of cycloxygenase-2 (COX-2), the formation of nitrotyrosine and the release of TNF-α as well as the activation of NF-κB in the rat intestinal epithelial cells [105]. In an oxidative stress model induced by hydrogen peroxide (H_2_O_2_) in an intestinal epithelial cell line, the *A. membranaceus* extract decreases the ROS levels, and increases the antioxidant cytoprotective factors expression and nuclear factor-like 2 (Nrf2) activation. In addition, AST showed antiinflammatory properties by suppressing the lipopolysaccharide-induced NF-κB pathway, thereby decreasing the expression of inducible nitric oxide synthase (iNOS) in the macrophage RAW264.7 [106]. Similarly, the total flavonoids extracted from *A. membranaceus* effectively inhibited the production of inflammatory mediators, such as NO and cytokine IL-1β, TNF-α, IL-6 and IFN-γ in lipopolysaccharide-stimulated RAW 264.7 macrophages in a dose-dependent manner, whereas they promoted the production of these inflammatory mediators in unstimulated macrophages [95].

According to the previous study, the flavonoids of *A. membranaceus* inhibit the PC12 neuronal cell injury induced by glutamate by increasing the antioxidant enzyme activities of superoxide dismutase and glutathione peroxidase [107]. In addition, the flavonoids of *A. membranaceus* also showed a high scavenging activity to 1,1-diphenyl-2-picrylhydrazyl (DPPH) radicals in the cell-free system. Moreover, the APS possessed good antioxidant properties in ferric-reducing antioxidant power (FRAP), hydroxyl radical (•OH), DPPH and superoxide radical (•O_2_^−^) scavenging capacity [108]. Therefore, *A. membranaceus* and its bioactive components could be developed as a novel antioxidant agent against inflammation in various diseases.

### 5.4. Antidiabetic Effects

Diabetes, characterized by elevated blood glucose, is a globally metabolic disorder disease, which could damage the kidneys, eyes, heart and gastric mucosa, and could also lead to coma and death without proper treatment [109,110]. It was reported that formononetin (7-hydroxy-4′-methoxyisoflavone) had potential to treat diabetic retinopathy by inhibiting the secretion of vascular endothelial growth factor (VEGF) in the HIF-1α/VEGF signaling pathway, and reducing the expressions of PHD, HIF-1α and VEGF proteins [111]. Liu et al. recently found that an APS of AERP (Mw: 2.01 × 10^3^ kDa) has hypoglycemic properties in a *db/db* diabetic mice model by reducing hyperglycemia and tissue impairment, and promoting cognitive function [61].

In a diabetic model of human umbilical vein endothelial cells (HUVEC), AS-IV effectively protected the HUVEC injury induced by high glucose by promoting the cell proliferation, and suppressing the apoptosis and inflammatory reactions in HUVEC, through the inhibition of the *c-Jun Nterminal kinase* (JNK) signaling pathway [112]. Zhang et al. studied the effect of the APS on diabetic nephropathy in streptozotocin-induced diabetic male Sprague-Dawley rats, and found that the APS not only decreased the concentration of blood glucose, microalbuminuria and plasma lipid, but improved renal function and reduced the ratio of kidney weight to body weight. In addition, the APS decreased the expression level of NF-κB in the renal cortex and raised the IκB mRNA level, which indicated that the APS has the potential for prevention and treatment of the progress of diabetic nephropathy [113].

### 5.5. Hepatoprotective Effects

As the largest solid organ in the human body, the liver plays important roles in drug metabolism, detoxification and the production of chemicals, and is easily injured by viral infection, metabolic disorder, overdose of toxin ingestion and immunological insult [114]. Modern pharmacological studies have proved that the bioactive ingredients derived from *A. membranaceus* are clinically beneficial for hepatoprotection. In a previous study, the authors found the AS-IV has the potential for the treatment of hepatic steatosis with activities in reducing lipid accumulation and insulin resistance in HepG2 cells. In addition, the AS-IV induces the phosphorylation of SREBP-1c at Ser372 in an AMPK-dependent manner in HepG2 cells [115]. Moreover, in an adipose dysfunction model induced by a high-fat diet (HFD) feeding in male ICR mice, AS-IV significantly reduced the accumulation of adipose cAMP by promoting Akt phosphorylation and combination with PDE3B. In addition, AS-IV inhibits the overproduction of hepatic glucose by decreasing the ectopic fat deposition in the liver [116].

Yan et al. investigated the hepatoprotective effect of APS on carbon tetrachloride (CCl_4_)-induced chronic liver injury in Sprague-Dawley rat models, and found that the APS reduced the serum levels of aspartate aminotransferase (AST), alanine aminotransferase (ALT) and alkaline phosphatase (ALP), and significantly increased the activities of catalase (CAT) and superoxide dismutase (SOD) in the liver [69]. Dang et al. evaluated the protective effects of APS on chronic hepatic injury induced by CCl4 in male Sprague-Dawley rats and found that APS treatment reduced the serum levels of total bilirubin (TBIL) and ALT, and increased the albumin level and SOD activity in the liver [117]. Therefore, the APS could effectively protect hepatocytes and prevent hepatic injury by limiting free radical production, elevating antioxidant enzyme activities and reducing lipid peroxidation.

### 5.6. Antiviral Effects

Many studies have comprehensively proven that *A. membranaceus* has the important property of antiviral activity. Influenza is an influenza-virus-caused acute respiratory infection disease, while *A. membranaceus* exhibited the obvious activity of anti-influenza virus in a previous study described by Liang et al. [118]. They found that *A. membranaceus* could effectively increase the survival rate of influenza-virus-infected Raw264.7 cells, which were mainly caused by the increase in SOD activity, the reduction in the malondialdehyde level, and the regulation of the TLR3 signaling pathway and cell proliferating cycle.

In addition, hepatitis B virus (HBV) remains the leading cause of acute and chronic hepatitis, cirrhosis and liver cancer, and the current treatment strategy of antivirus drugs (such as lamivudine and interferon) is associated with various drawbacks, including low antiviral potency, side effects and a long treatment period [119]. Therefore, Dang et al. evaluated the inhibitory effects of the APS on HBV replication in HBV transgenic mice [120]. They reported that the administration of APS and emodin decreased the viral DNA levels in the serum, and reduced the serum levels of the surface antigen of hepatitis B virus (HBsAg), hepatitis B e antigen (HBeAg) and hepatitis B core antigen (HBcAg) in the mice, which indicated that the APS and emodin had a persistent inhibitory effect on virus replication in vivo. Du et al. reported that APS effectively increased the T cells’ proliferating activity and improved the HBsAg-related antibody level, thus inhibiting the viral reproduction by inducing the CD4^+^ and CD8^+^ T cells to produce cytokines and protecting the body against viruses [121]. In a human liver cell line, HepG2 2.2.15 transfected by HBV, the bioactive ingredient of AS-IV effectively inhibits the secretion of HBsAg and HBeAg, which suggests that AS-IV has potent anti-HBV activity in vitro and deserves to be further developed as an anti-HBV agent [13].

### 5.7. Anti-Cardiovascular Effects

Cardiovascular diseases are mainly caused by diabetes, hypertension, high cholesterol and dyslipidemia syndrome, and the most notable symptoms are ischemic or hemorrhagic lesions in the heart, brain and whole body [12]. Wang et al. investigated the effects of *Astragalus* flavonoids on cardiovascular disease in vivo [122]. In a diet-induced atherosclerotic rabbit model, the flavonoids significantly decreased the total cholesterol levels in plasma, reduced the aortic fatty streak area, and effectively scavenged the hydroxyl radicals and superoxide in a concentration-dependent manner.

A study was carried out by Wu et al. to explore the vasorelaxation effects of formononetin on an isolated rat aorta and underlying mechanisms [123]. Formononetin possesses vascular relaxation in a endothelium-independent and endothelium-/NO-dependent manner by activating the adenosine triphosphate (ATP)-dependent K^+^ (K_ATP_) and Ca^2+^-activated K^+^ (BK_Ca_) channels. Similarly, calycosin has proven to be an excellent endothelium-independent vasorelaxant on precontracted thoracic aortic rings in rats due to its function of being a noncompetitive Ca^2+^ channel blocker [124]. In addition, Zhu et al. evaluated the effects of formononetin derivative (sodium formononetin-3′-sulphonate) on angiogenesis and neuroprotection in a cerebral ischemia and reperfusion injury rat model [125]. They found that sodium formononetin-3′-sulphonate effectively protected the brain from the ischemia and reperfusion injury in vivo, which caused the improvement in neurological function, and an increase in the expression of the vascular endothelial growth factor and platelet endothelial cell adhesion molecule, as well as the suppression of cell apoptosis.

### 5.8. Toxicity and Clinical Trials

A few studies on the preclinical toxicity of bioactive compounds in *A. membranaceus* were carried out to verify the clinical safety. It was reported that the AS-IV has fetal toxicity at a dose more than 0.5 mg/kg and maternal toxicity after intravenous administration of 1.0 mg/kg, while it has no teratogenic effects in rats and rabbits [126]. In addition, the AS-IV delayed the fur development, liff parry reflex, and eye opening after birth under 1.0 mg/kg during a reproductive toxicity test in Sprague-Dawley rats, while there was no effect on the memory and learning [127].

In addition, it is necessary to evaluate the therapeutic effects of *A. membranaceus* on humans in clinical trials. A positive effect in patients with myocardial infarction was reported following the administration of Tongguan Capsules (TGC, composed of: *A. membranaceus*, *Borneolum syntheticum*, *Salvia miltiorrhiza and Grasshopper*), at a dose of 4.5 g/day for 6 months, who demonstrated a reduced left ventricular end-systolic volume index as well as decreased myocardial markers of fibrosis and apoptosis and reduced circulating levels of inflammatory cytokines [128]. Lee et al. reported a study in children with growth retardation syndrome who received HT042, containing a mixture of *A. membranaceus* roots, *Eleutherococcus senticosus* stems and *Phlomis umbrosa* roots, twice a day for 24 weeks; the HT042-receiving patients showed a significant increase in insulin-like growth factor binding protein-3 (IGFBP-3) and IGF-1, as well as an increase in the height and weight of children [129].

## 6. Future Perspectives

### 6.1. Future Market Prospects of A. membranaceus

With the improvement in people’s living standards and emphasis on health, the demand for *A. membranaceus* has increased due to its vital role in the prevention and early interventional treatment of diseases. It is worth noting that HSBD (Huashibaidu granules, *A. membranaceus* as a major component) was clinically proven to be effective for the treatment of COVID-19 patients [130,131]. Thus, the therapeutic effect of *A. membranaceus* on COVID-19 has promoted the sharp increase in its market price. There is expected to be an increasing demand for *A. membranaceus* in the future market following the worldwide spread of the epidemic. Nevertheless, the *A. membranaceus* industry has a long chain with a wide range, including cultivation, processing, acquisition, storage, transportation, as well as product research and development. Moreover, the *A. membranaceus* industry is more oriented towards quality assurance than production yield. Therefore, it is imperative to promote the establishment of price formation mechanisms oriented by the quality and to alleviate the large fluctuation of market prices and the total output caused by production dispersion, as well as strengthen the guidance practices on the development of the *A. membranaceus* industry.

### 6.2. Development Trend of Biotechnology in A. membranaceus

At the current stage, the numerous studies on the biosynthesis of flavonoids and triterpene saponins, and the structural analysis of polysaccharides in *A. membranaceus* have mainly focused on the discovery of important structural and regulatory genes involved in the biosynthetic pathway, while the key enzyme functions associated with the synthesis of these bioactive ingredients in *A. membranaceus* are still unknown, especially those involved in structural modification, transcriptional regulation and the catalytic process, thereby restricting the industrialized production and sustainable supply of natural products with pharmacological activities.

Therefore, it is imperative to comprehensively understand the biosynthetic pathway and regulatory mechanism of bioactive ingredients, and an effective way for further addressing these bottlenecks is heterologous biosynthesis by using heterologous plants or microorganisms as the chassis of cell factories that have transformed the metabolic flux and reconstructed the biosynthetic pathway of bioactive ingredients. Moreover, it is currently still a challenge to improve the accumulation of target products and realize the efficient and large-scale production by optimizing the chassis cell system, the regulatory factors and the fermentation conditions on the basis of heterologous biosynthesis. At present, there has not been much study on the synthetic biology in *A. membranaceus*, which still needs more in-depth exploration and breakthroughs in the future.

## 7. Conclusions

With increasing demand for traditional Chinese medicine, the sustainable development of the *A. membranaceus* industry has received more attention. Thus, this review focuses primarily on the biosynthesis pathway of flavonoids and triterpenoid saponins, as well as the structural features of polysaccharides derived from *A. membranaceus*. Nevertheless, the biosynthetic processes of flavonoids and astragaloside in *A. membranaceus* have not been completely resolved, and there is a lack of key steps in the synthesis pathways, which thwarts the large-scale production of bioactive ingredients. In addition, the pharmacological activities of these bioactive components were also summarized, which provided a more comprehensive understanding for the traditional Chinese medicine development and clinical applications. Finally, we also discussed the future market prospects and development trend of the bioengineering technology of *A. membranaceus*, hoping to lay a foundation for the in-depth study and utilization of *A. membranaceus*.

## Figures and Tables

**Figure 1 molecules-28-05018-f001:**
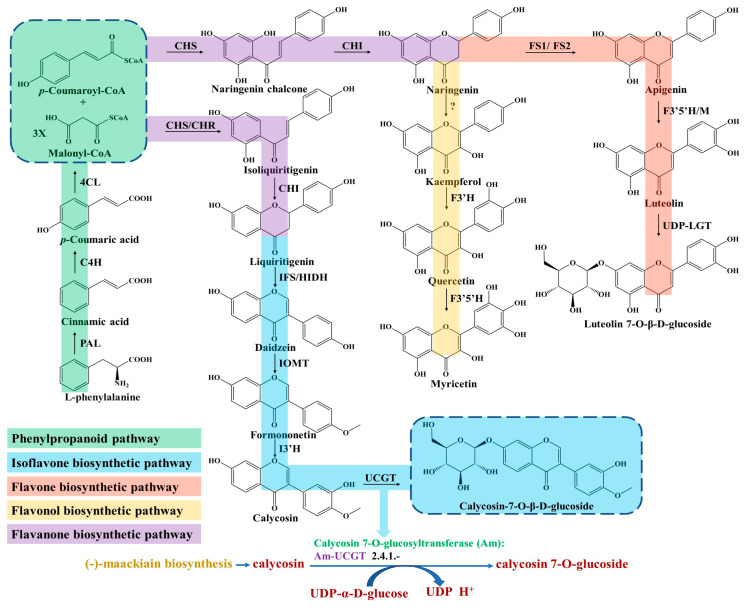
Biosynthesis pathway of flavonoids in *A. membranaceus*.

**Figure 2 molecules-28-05018-f002:**
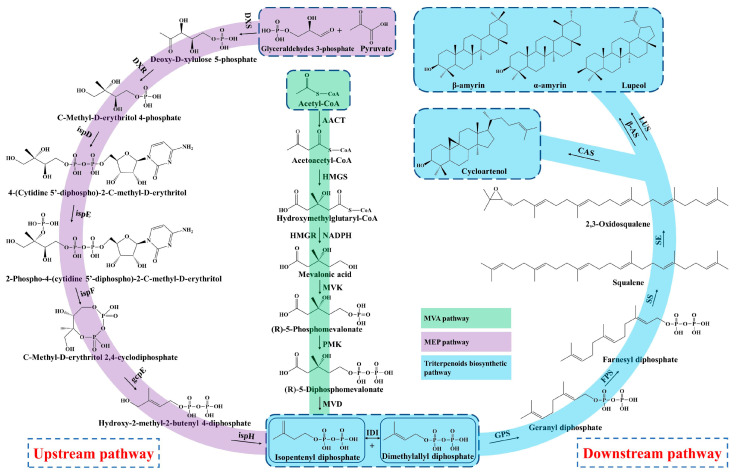
Biosynthesis pathway of triterpenoid saponins in *A. membranaceus*.

**Figure 3 molecules-28-05018-f003:**
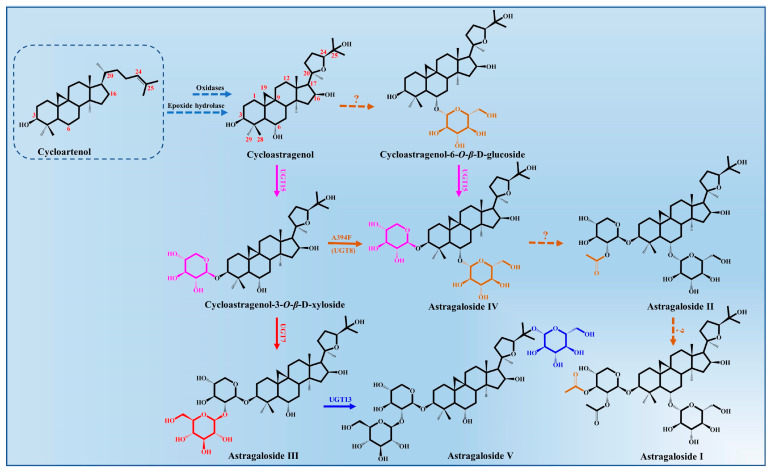
Astragalosides’ biosynthesis pathways from cycloartenol in *A. membranaceus*.

**Figure 4 molecules-28-05018-f004:**
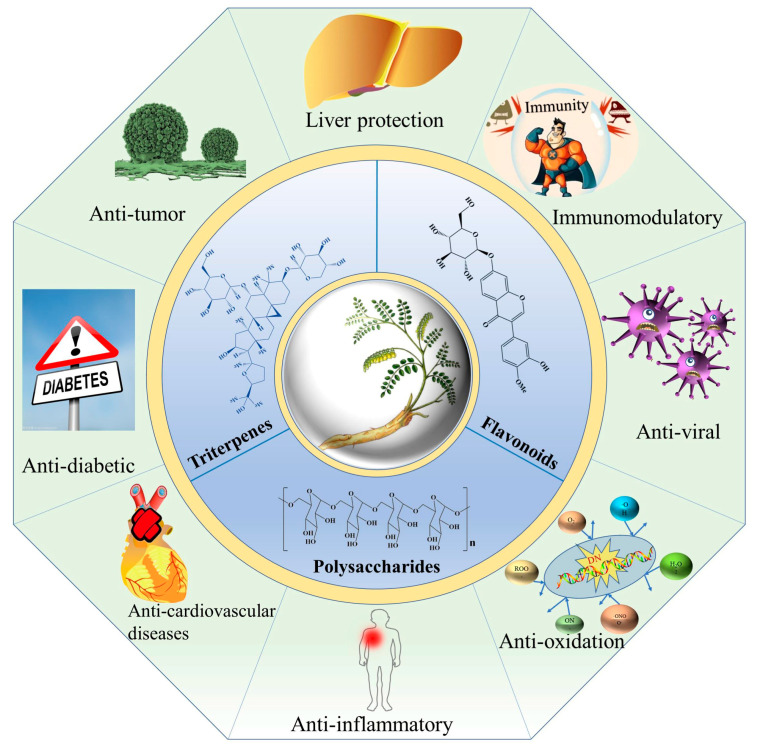
Pharmacological activities of flavonoids, triterpene saponins and polysaccharides derived from *A. membranaceus*.

**Figure 5 molecules-28-05018-f005:**
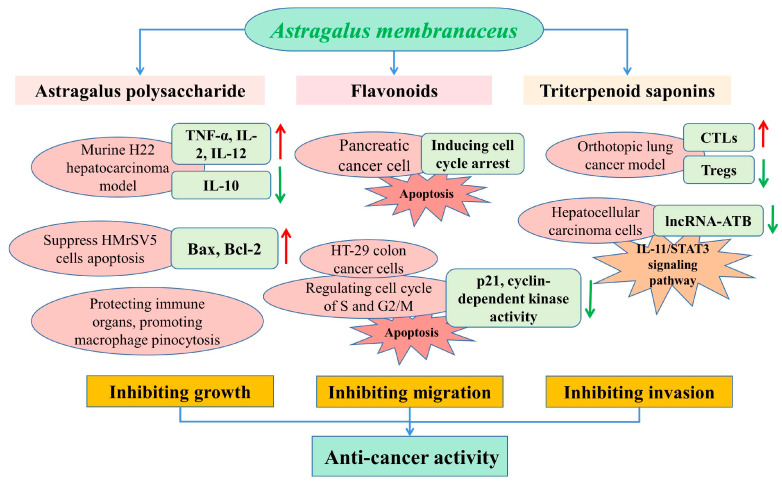
Schematic presentation of the anticancer mechanisms of main bioactive compounds in *A. membranaceus*.

**Table 2 molecules-28-05018-t002:** Comparison of pharmacological potentials between *A. membranaceus* and other *Astragalus* genus plants.

Species Resource	Main Bioactive Compounds	Potential Pharmacological Activities	References
*A. membranaceus*	Flavonoids, triterpene saponins, polysaccharides	Anticancer, antidiabetic, antiviral, hepatoprotective, immunomodulatory, antiinflammatory, antioxidant, anti-cardiovascular activities.	[75]
*A. abyssinicus*	Flavonoids	Antioxidant	[76]
*A. monspessulanus*	Flavonoids	Antioxidant	[77]
*A. hoantchy*	Flavonoids	Antibacterial	[78]
*A. adsurgens*	Flavonoids	Antiinflammatory	[79]
*A. brachycalyx*	Triterpene saponins	Immunomodulatory	[80]
*A. plumosus*	Triterpene saponins	Antioxidant	[81]
*A. boeticus*	Triterpene saponins	Anticancer	[82]
*A. pennatulus*	Triterpene saponins	-	[83]

## Data Availability

Not applicable.

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
