# Peer review of "Biosynthesis and Pharmacological Activities of Flavonoids, Triterpene Saponins and Polysaccharides Derived from Astragalus membranaceus"

_molecules, 2023, doi:10.3390/molecules28135018_

Round 1

Reviewer 1 Report

I have reviewed the manuscript entitled "Biosynthesis and pharmacological activities of flavonoids, triterpene saponins and polysaccharides derived from Astragalus membranaceus".

This is an exhaustive review of the main bioactive compounds present in A. membranaceus (flavonoids, triterpene saponins, and polysaccharides), their biosynthesis, and their action against different diseases.

The manuscript is well written, easy to read and follow. Minor typos that can be corrected in the editing process.

This manuscript contributes to a better understanding of the biological processes involved in the production of these biocompounds, highlighting the enzymes and genes responsible for each pathway and providing target genes for the possible elicitation of these compounds in A. menbranaceus thus increasing their production.

Some questions that arise to me when reading the manuscript are:

Wild A. menbranaceus is an endangered species as the authors state, as this plant is widely used in traditional Chinese medicine, there should be an orchard to produce large amounts of A. menbranaceus, right? What are the economic benefits of producing these compounds by heterologous biosynthesis instead of extracting them from plants? What does heterologous biosynthesis mean?

In the abstract the authors state "we propose potential heterologous biosynthetic strategies to enhance the industrialized production and sustainable supply of natural products with pharmacological activities in A. membranaceus." What are those strategies? What has already been done about heterologous biosynthesis in A. membranaceus? I believe that this review is valuable not only for heterologous biosynthesis, but also for the knowledge of the synthesis of these biocompounds in A. membranaceus for their elicitation in the whole plant, for their production in cell culture biofactories, etc.

I recommend the publication of this manuscript.

Author Response

Comment 1: I have reviewed the manuscript entitled "Biosynthesis and pharmacological activities of flavonoids, triterpene saponins and polysaccharides derived from Astragalus membranaceus". This is an exhaustive review of the main bioactive compounds present in A. membranaceus (flavonoids, triterpene saponins, and polysaccharides), their biosynthesis, and their action against different diseases. The manuscript is well written, easy to read and follow.

Minor typos that can be corrected in the editing process.

This manuscript contributes to a better understanding of the biological processes involved in the production of these biocompounds, highlighting the enzymes and genes responsible for each pathway and providing target genes for the possible elicitation of these compounds in A. menbranaceus thus increasing their production.

Response 1: First of all, thank you very much for your recognition. Your recognition motivated us greatly to continue our research. We have tried our best to improve the manuscript according to your valuable comments and suggestions. We hope that the current modification and correction in the revised manuscript is satisfactory.

Some questions that arise to me when reading the manuscript are:

Comment 2: Wild A. menbranaceus is an endangered species as the authors state, as this plant is widely used in traditional Chinese medicine, there should be an orchard to produce large amounts of A. menbranaceus, right? What are the economic benefits of producing these compounds by heterologous biosynthesis instead of extracting them from plants? What does heterologous biosynthesis mean?

Response 2: We would like to thank you for your questions. By the 1980s, the wild resources of A. menbranaceus were in short supply due to the overexploitation, which led to the emergence of the much needed cultivated A. menbranaceus. The cultivated A. menbranaceus is primarily planted on farmland with sandy loam where the soil is loose with a thickness of approximately 60-90 cm and grown for 2 years but cannot exceed 3 years. To date, although the use of heterologous biosynthesis to produce natural products with medicinal value has received increasing attention, it is still a challenging task to achieve an industrialized yield of target products due to its long cycle, high cost and low efficiency. The heterologous biosynthesis is the production of target products with medicinal value by using the heterologous hosts, such heterologous plants, E. coli and S. cerevisiae

Comment 3: In the abstract the authors state "we propose potential heterologous biosynthetic strategies to enhance the industrialized production and sustainable supply of natural products with pharmacological activities in A. membranaceus." What are those strategies? What has already been done about heterologous biosynthesis in A. membranaceus?

Response 3: Thank you for your helpful comments. These strategies are heterologous biosynthesis by using heterologous plants or microorganisms (E. coli, S. cerevisiae, etc.) as the chassis of cell factories that transformed the metabolic flux and/or reconstructed the biosynthetic pathway of bioactive ingredients of A. membranaceus, such as calycosin-7-O-β-D-glucoside, calycosin, astragalosides. Although the heterologous biosynthesis of bioactive ingredients in A. membranaceus  could increase the output of target natural products, this method also suffers from many drawbacks such as long cycle, high cost and low efficiency. Thus, there were few studies about heterologous biosynthesis of bioactive ingredients in A. membranaceus.

Comment 4: I believe that this review is valuable not only for heterologous biosynthesis, but also for the knowledge of the synthesis of these biocompounds in A. membranaceus for their elicitation in the whole plant, for their production in cell culture biofactories, etc. I recommend the publication of this manuscript.

Response 4: Once again, thank you very much for your recognition and recommendation.

Reviewer 2 Report

Title - Biosynthesis and pharmacological activities of flavonoids, tritepene saponins and polysaccharides derived from Astragalus membranaceus

Flavonoids, triterpene saponins, and polysaccharides, which are the most important bioactive components of A. membranaceus, were chosen for discussion in this study. The phenylpropanoid route is first used to synthesize the flavonoids, and then the process is catalyzed by suitable enzymes. While the standard upstream pathways of mevalonate (MVA) and methylerythritol phosphate (MEP) and the downstream pathways of triterpenoid skeleton generation and modulation are used to produce triterpenoid saponins, mainly astragalosides. These substances are essential for liver protection, immunological modulation, anti-cancer, anti-diabetic, anti-viral, anti-inflammatory, and anti-cardiovascular activities and have a wide spectrum of biological and pharmacological effects. It is beneficial to discuss the numerous health advantages of bioactive chemicals found in herbal plants. In the pharmaceutical industry, it will promote the use of Astragalus membranaceus. Nevertheless, it is crucial to consider current market trends along with comparison of other relevant sources that are known to provide similar bioactive chemicals.

Comments

Line 16-19 “The main bioactive compounds of flavonoids, triterpene saponins, and polysaccharides of A. membranaceus showed a wide range of biological activities and pharmacological effects, and played an important role in protecting the liver, immunomodulation, anti-cancer, anti-diabetic, anti-viral, anti-inflammatory, anti-oxidant and anti-cardiovascular activities.”  This sentence have major grammatical error making sentence sense less .Replace this line with “The main bioactive compounds including flavonoids, triterpene saponins, and polysaccharides obtained from A. membranaceus showed a wide range of biological activities and pharmacological effects. These bioactive compounds have significant role in protecting the liver, immunomodulation, anti-cancer, anti-diabetic, anti-viral, anti-inflammatory, anti-oxidant and anti-cardiovascular activities”.

Figure 1 Text in figure 1 is blur increase bold it

Either use Astragalus membranaceus or A. membranaceus  in entire manuscript

Line 21 no need of space between 20 000 instead add comma

Line 145, 234 What is and/or means?

Line 202 “AmOSC2” Why it is italicized?

Table 1 Rewrite table tile Its incomplete and make it self-explanatory. The orientations of headings is inappropriate improve it.

Figure 1 contain some content in another language add it in English language

Line 364 Repetition of word “in” rewrite sentence and why in vivo italicized

Line 388 Write complete name of “Astragalus”. Why it is not italicized

Start heading from fresh page not last line like section 3.2, 5.5

Line 428-430 Sentence have word in different color make them black.

There are lots of text in section 5.5, 5.6 and 5.7 that are not black color revise it.

There are lots of old citations, avoid citations older than 20th century. Update the reference list

Provide comparison of pharmacological potentials of A. membranaceus with other relevant species. You can also add a table of comparison to highlight the significance of A. membranaceus in medicinal sector.

What is the gap of knowledge indicated in the paper is not clear. Put some light on gap of knowledge

Add new section about the products currently available in market along with companies, manufactures, products and cost to explore market trends

In section 5.2 Anticancer effect lots of mechanisms discussed add a flowchart/ table /diagram on mechanism involved along with type of cancer or cell lines on which anticancer activity is reported using A. membranaceus bioactive compounds.

To make manuscript appealing commercial section must be added that will enhance the impact of review. There are multiple grammatical and editing error need to addressed

Future perspective sounds incomplete as it did not discuss about any figures related to the production of aforementioned bioactive compounds or commercial values in upcoming decade.

It is important to add conclusion at the end before reference to briefly describes the outcomes obtained from this manuscript.

 The bioactive compounds obtained from A. membranaceus are currently obtained from microorganism as well such as microalgae. Add some text for this as advantage or challenge before future perspective.

Latest References are lacking. Only one reference for the year 2023. Cite more references of the recent year.

Moderate editing of English language is required. Typo errors must be removed that is existing at many places in the article. 

Author Response

Title - Biosynthesis and pharmacological activities of flavonoids, tritepene saponins and polysaccharides derived from Astragalus membranaceus

Flavonoids, triterpene saponins, and polysaccharides, which are the most important bioactive components of A. membranaceus, were chosen for discussion in this study. The phenylpropanoid route is first used to synthesize the flavonoids, and then the process is catalyzed by suitable enzymes. While the standard upstream pathways of mevalonate (MVA) and methylerythritol phosphate (MEP) and the downstream pathways of triterpenoid skeleton generation and modulation are used to produce triterpenoid saponins, mainly astragalosides. These substances are essential for liver protection, immunological modulation, anti-cancer, anti-diabetic, anti-viral, anti-inflammatory, and anti-cardiovascular activities and have a wide spectrum of biological and pharmacological effects. It is beneficial to discuss the numerous health advantages of bioactive chemicals found in herbal plants. In the pharmaceutical industry, it will promote the use of Astragalus membranaceus. Nevertheless, it is crucial to consider current market trends along with comparison of other relevant sources that are known to provide similar bioactive chemicals.

Thank you very much for your helpful comment. The current market trends along with comparison of other relevant sources have been added in the revised manuscript and the detailed improvement were presented in “Response 14”, “Response 16” and “Response 18”.

Comment 1: Line 16-19 “The main bioactive compounds of flavonoids, triterpene saponins, and polysaccharides of A. membranaceus showed a wide range of biological activities and pharmacological effects, and played an important role in protecting the liver, immunomodulation, anti-cancer, anti-diabetic, anti-viral, anti-inflammatory, anti-oxidant and anti-cardiovascular activities.” This sentence have major grammatical error making sentence sense less. Replace this line with “The main bioactive compounds including flavonoids, triterpene saponins, and polysaccharides obtained from A. membranaceus showed a wide range of biological activities and pharmacological effects. These bioactive compounds have significant role in protecting the liver, immunomodulation, anti-cancer, anti-diabetic, anti-viral, anti-inflammatory, anti-oxidant and anti-cardiovascular activities”.

Response 1: Thank you for your careful checking. According to your suggestion, we have revised the “The main bioactive compounds of flavonoids, triterpene saponins, and polysaccharides of A. membranaceus showed a wide range of biological activities and pharmacological effects, and played an important role in protecting the liver, immunomodulation, anti-cancer, anti-diabetic, anti-viral, anti-inflammatory, anti-oxidant and anti-cardiovascular activities.” into “The main bioactive compounds including flavonoids, triterpene saponins, and polysaccharides obtained from A. membranaceus showed a wide range of biological activities and pharmacological effects. These bioactive compounds have significant role in protecting the liver, immunomodulation, anti-cancer, anti-diabetic, anti-viral, anti-inflammatory, anti-oxidant and anti-cardiovascular activities.”. Changes have been marked in red color on page 1 from line 17-21.

Comment 2: Figure 1 Text in figure 1 is blur increase bold it

Response 2: We greatly appreciate you for pointing out this problem. We have increased the bold of text in figure 1 and ensured that the current version are clear.

Comment 3: Either use Astragalus membranaceus or A. membranaceus in entire manuscript

Response 3: Thank you for your helpful comments. The full name of Astragalus membranaceus was used in manuscript for the first appearance, followed by the abbreviation of A. membranaceus.

Comment 4: Line 21 no need of space between 20 000 instead add comma

Response 4: Thank you for your elaborative checking. We have added the comma between 20 000 in the revised manuscript and marked in red color on page 4 line 46.

Comment 5: Line 145, 234 What is and/or means?

Response 5: Thank you for your comments. Based on the literature review, we have revised the “and/or” into “or” in the revised manuscript. Changes have been marked in red color.

Comment 6: Line 202 “AmOSC2” Why it is italicized?

Response 6: Thank you for your question. The“AmOSC2”is a name of gene, thus it is italicized.

Comment 7: Table 1 Rewrite table tile Its incomplete and make it self-explanatory. The orientations of headings is inappropriate improve it.

Response 7: Thank you for your helpful comments. We have rewritten the table title as “Table 1. The chemical composition, structural properties and pharmacological activities of polysaccharides derived from the A. membranaceus.” in the revised manuscript. Changes have been marked in red color on page 9 from line 293-294.

Comment 8: Figure 1 contain some content in another language add it in English language

Response 8: We greatly appreciate you for pointing out this problem. We have replaced the Chinese in English in Figure 4.

Comment 9: Line 364 Repetition of word“in”rewrite sentence and why in vivo italicized

Response 9: Thank you for your careful checking. The repetition of word “in” have been deleted in the revised manuscript. In addition, the “in vivo” is derived from Latin, thus it is italicized.

Comment 10: Line 388 Write complete name of“Astragalus”. Why it is not italicized

Response 10: Thank you for your comments. The complete name of “A. membranaceus” have been supplemented and it was italicized in the revised manuscript. Changes have been marked in red color.

Comment 11: Start heading from fresh page not last line like section 3.2, 5.5

Response 11: We would like to thank you for your helpful suggestion. According to your suggestion, we have started the heading from fresh page in the revised manuscript.

Comment 12: Line 428-430 Sentence have word in different color make them black. There are lots of text in section 5.5, 5.6 and 5.7 that are not black color revise it.

Response 12: Thank you for your helpful comments. We have revised the color of text into black in the revised manuscript.

Comment 13: There are lots of old citations, avoid citations older than 20th century. Update the reference list

Response 13: Thank you for your valuable comments. The reference list has been updated in the revised manuscript. Changes have been marked in red color.

Comment 14: Provide comparison of pharmacological potentials of A. membranaceus with other relevant species. You can also add a table of comparison to highlight the significance of A. membranaceus in medicinal sector.

Response 14: Thank you for your helpful suggestions. According to your suggestion, the “Table 2. of Comparison of pharmacological potentials between A. membranaceus and other Astragalus genus plants” has been added in the revised manuscript. Changes have been marked in red color on page 11 in line 303.

Comment 15: What is the gap of knowledge indicated in the paper is not clear. Put some light on gap of knowledge

Response 15: Many thanks for your valuable comments. The gap of knowledge indicated in the paper have been supplemented in the revised manuscript and list below. Changes have been marked in red color on page 18 from line 557-561, on page 19 from line 566-569, on page 19 from line 577-579.

“While the key enzymes functions associated with synthesis of these bioactive ingredients in A. membranaceus are still unknown, especially involved in structural modification, transcriptional regulation, and catalytic process, thereby restricting the industrialized production and sustainable supply of natural products with pharmacological activities.” “Moreover, it is currently still a challenge to improve the accumulation of target products and realize the efficient and large-scale production by optimizing the chassis cell system, regulatory factors and fermentation conditions on the basis of heterologous biosynthesis.” “Nevertheless, the biosynthetic process of flavonoids and astragaloside in A. membranaceus have not been completely resolved, and there is a lack of key steps in synthesis pathways, which thwarts the large-scale production of bioactive ingredients.”

Comment 16: Add new section about the products currently available in market along with companies, manufactures, products and cost to explore market trends

Response 16: Thank you for your helpful suggestions. According to your suggestion, a new section of “6.1 Future market prospects of A. membranaceus” has been added in the revised manuscript. Changes have been marked in red color on page 18 from line 537-552.

Comment 17: In section 5.2 Anticancer effect lots of mechanisms discussed add a flowchart/ table /diagram on mechanism involved along with type of cancer or cell lines on which anticancer activity is reported using A. membranaceus bioactive compounds.

Response 17: Thank you for your helpful comments. The schematic presentation of the anti-cancer mechanisms of main bioactive compounds in A. membranaceus has been supplemented in the revised manuscript (Fig.5) and listed below. Changes have been marked in red color.

Figure 5. Schematic presentation of the anti-cancer mechanisms of main bioactive compounds in A. membranaceus.

Comment 18: To make manuscript appealing commercial section must be added that will enhance the impact of review. There are multiple grammatical and editing error need to addressed

Response 18: Really appreciate your valuable comments. The commercial section has been also added in the “6.1 Future market prospects of A. membranaceus” in the revised manuscript. In addition, we have had a double check of our manuscript about the grammar or use of English and made improvements in the revised manuscript. Changes have been marked in red color.

Comment 19: Future perspective sounds incomplete as it did not discuss about any figures related to the production of aforementioned bioactive compounds or commercial values in upcoming decade.

Response 19: We are grateful for your comments. The “6. Future perspectives” section has been carefully revised and added the section of “6.1 Future market prospects of A. membranaceus” and “6.2 Development trend of biotechnology in A. membranaceus” in the revised manuscript. Changes have been marked in red color.

Comment 20: It is important to add conclusion at the end before reference to briefly describes the outcomes obtained from this manuscript.

Response 20: We would like to thank you for your valuable suggestion. According to your suggestion, the section of “Conclusions” has been added in the revised manuscript. Changes have been marked in red color on page 19 from line 572-585.

Comment 21: The bioactive compounds obtained from A. membranaceus are currently obtained from microorganism as well such as microalgae. Add some text for this as advantage or challenge before future perspective.

Response 21: Many thanks for your valuable comments. The bioactive compounds obtained from A. membranaceus are different from microorganism as well such as microalgae. Nevertheless, the bioactive compounds derived from A. membranaceus  could be biosynthesized by using heterologous microorganisms as the chassis of cell factories that transformed the metabolic flux and reconstructed the biosynthetic pathway of bioactive ingredients. Therefore, the challenge of heterologous biosynthesis of bioactive ingredients by microorganisms has been added in the section of “6.2 Development trend of biotechnology in A. membranaceus” in the revised manuscript. Changes have been marked in red color on page 19 from line 562-571.

Comment 22: Latest References are lacking. Only one reference for the year 2023. Cite more references of the recent year.

Response 22: Thank you for your valuable comments. The reference list has been updated, and the references of the recent years have been cited in the revised manuscript. Changes have been marked in red color.

Comment 23: Moderate editing of English language is required. Typo errors must be removed that is existing at many places in the article. 

Response 23: We would like to thank you for your helpful comments. According to your comments, we have had a double check of our manuscript about the grammar or typo errors and made improvements in the revised manuscript. Changes have been marked in red color.

Reviewer 3 Report

The Review deal with biosynthesis and pharmacological activities of the main bioactive compounds of flavonoids, triterpene saponins and polysaccharides of A. membranaceus.

In the manuscript there are many mistakes:

Lines 61/62: the sentence is incomplete.

Line 117: anthoanidins

Line 183: the reference Duan et al. missing of the year. Am must in Italics.

Lines 295 and others: what do you mean with ASP? Do you intend APS.

The text doesn’t report Fig1.

Lines 490-494: the sentence is not clear.

The Table 1 is not easily understood.

The manuscript is very long in the description of the biological activities of different compounds, it looks a list of different results.

In the future perspectives you do not describe well potential strategies…. (Lines 28-31), which are mentioned in the abstract and introduction sections. It would be good to add a conclusion.

The English quality is poor.

Author Response

The Review deal with biosynthesis and pharmacological activities of the main bioactive compounds of flavonoids, triterpene saponins and polysaccharides of A. membranaceus.

In the manuscript there are many mistakes:

Comment 1: Lines 61/62: the sentence is incomplete.

Response 1: Thank you for your kind reminding. The sentence has been revised as “In nature, triterpenes and their saponins are the second largest secondary metabolites with a total of more than 20,000, which widely distributed in higher plants, dicotyledons, monocotyledons, fungi, pteridophytes and marine organisms”. Changes have been marked in red color.

Comment 2: Line 117: anthoanidins

Response 2: Thank you for your elaborative checking. The “anthoanidins” has been removed in the revised manuscript.

Comment 3: Line 183: the reference Duan et al. missing of the year. Am must in Italics.

Response 3: Thank you very much for your helpful comment. The year of the reference Duan et al. has been supplemented, and the Am has been italicized in the revised manuscript. Changes have been marked in red color.

Comment 4: Lines 295 and others: what do you mean with ASP? Do you intend APS.

Response 4: Thank you for your careful checking. The “ASP” was a mistake and it has been corrected as “APS” in revised manuscript. Changes have been marked in red color in the revised manuscript.

Comment 5: The text doesn’t report Fig1.

Response 5: Thank you very much for your helpful comment. In the revised manuscript, the Fig1 has been reported and changes have been marked in red color.

Comment 6: Lines 490-494: the sentence is not clear.

Response 6: Really appreciate your valuable comments. The sentence has been revised as “They found that sodium formononetin-3′-sulphonate could effectively protect the brain from the ischemia and reperfusion injury in vivo, which caused by the improving neurological function, increasing the expression of vascular endothelial growth factor and platelet endothelial cell adhesion molecule, as well as suppressing the cell apoptosis.” Changes have been marked in red color on page 7 from line 1-7.

Comment 7: The Table 1 is not easily understood.

 Response 7: Thank you for your helpful comments. We have optimized the Table 1 contents and rewritten the table title as “Table 1. The chemical composition, structural properties and pharmacological activities of polysaccharides derived from the A. membranaceus.” in the revised manuscript. Changes have been marked in red color.

Comment 8: The manuscript is very long in the description of the biological activities of different compounds, it looks a list of different results.

Response 8: We really appreciate your helpful comment. In order to systematically summarize the pharmacological effects of bioactive ingredients in A. membranaceus,  a large number of relative results have been cited in the manuscript. In addition, the schematic presentation of the anti-cancer mechanisms of main bioactive compounds in A. membranaceus has been supplemented in the revised manuscript (Fig.5). Changes have been marked in red color.

Comment 9: In the future perspectives you do not describe well potential strategies…. (Lines 28-31), which are mentioned in the abstract and introduction sections. It would be good to add a conclusion.

Response 9: Many thanks for your valuable comments. The “6. Future perspectives” section has been carefully revised and added the section of “6.1 Future market prospects of A. membranaceus” and “6.2 Development trend of biotechnology in A. membranaceus” in the revised manuscript. In addition, the section of “Conclusions” has been added in the revised manuscript. Changes have been marked in red color on page 18 from line 537-571.

Comment 10: The English quality is poor.

Response 10: Many thanks for your valuable comments. We have had a double check of our manuscript about the grammar or typo errors and made improvements in the revised manuscript. Changes have been marked in red color.

Reviewer 4 Report

The manuscript was very well written.
It contains all the necessary aspects needed to properly discuss the topic.
It focuses on the biochemical changes taking place in Astragalus membranaceus,
as well as the aspect of its effect on the human body.
Authors are being ask to reconsider to change the description in the figure 4.
There is "liver production" - in my opinion it wlii be much more proper to write "liver protection".
I think that
it would be valuable to add a few sentences (a paragraph) regarding possible
adverse changes/interactions that may occur in the human body.
Most of recerences relate to animal study and what about human study,
and what is more human safety?

Author Response

Comment 1: The manuscript was very well written. It contains all the necessary aspects needed to properly discuss the topic. It focuses on the biochemical changes taking place in Astragalus membranaceus, as well as the aspect of its effect on the human body.

Response 1: Thank you very much for your recognition. Your recognition motivated us greatly to continue our research. We have tried our best to improve the manuscript according to your valuable comments and suggestions.

Comment 2: Authors are being ask to reconsider to change the description in the figure 4. There is "liver production" - in my opinion it will be much more proper to write "liver protection".

Response 2: Thank you for pointing out this problem. We have revised “liver production” into “liver protection” in Figure 4 in the revised manuscript.
Comment 3: I think that it would be valuable to add a few sentences (a paragraph) regarding possible adverse changes/interactions that may occur in the human body. Most of references relate to animal study and what about human study, and what is more human safety?

Response 3: We would like to thank you for your helpful suggestion. According to your suggestion, a new section of “5.8. Toxicity and clinical trials” has been supplemented in the revised manuscript. Changes have been marked in red color on page 17 from line 516-535.

Round 2

Reviewer 2 Report

Authors answered the queries and modified the manuscript.